# Performance and resource requirements of in-person versus voice call versus automated telephone-based socioeconomic data collection modalities for community-based health programmes: a systematic review protocol

Luke Nelson Allen  ,[1] Shona Mackinnon,[2] Iris Gordon,[1] David Blane  ,[2] Ana Patricia Marques  ,[3] Stephen Gichuhi,[4] Alice Mwangi,[5] Matthew J Burton,[1] Nigel Bolster,[3,6] David Macleod,[3,7] Min Kim,[8] Jacqueline Ramke  ,[1] Andrew Bastawrous[3,7]

For numbered affiliations see end of article.

**Correspondence to**
Dr Luke Nelson Allen;
drlukeallen@gmail.com

## ABSTRACT

**Introduction** Gathering data on socioeconomic status (SES) is a prerequisite for any health programme that aims to assess and improve the equitable distribution of its outcomes. Many different modalities can be used to collect SES data, ranging from (1) face-to-face elicitation, to (2) telephone-administered questionnaires, to (3) automated text message-based systems. The relative costs and perceived benefits to patients and providers of these different data collection approaches is unknown. This protocol is for a systematic review that aims to compare the resource requirements, performance characteristics, and acceptability to participants and service providers of these three approaches to collect SES data from those enrolled in health programmes.

**Methods and analysis** An information specialist will conduct searches on the Cochrane Library, MEDLINE, Embase, Global Health, ClinicalTrials.gov, the WHO ICTRP and OpenGrey. All databases will be searched from 1999 to present with no language limits used. We will also search Google Scholar and check the reference lists of relevant articles for further potentially eligible studies. Any empirical study design will be eligible if it compares two or more modalities to elicit SES data from the following three; in-person, voice call, or automated phone-based systems. Two reviewers will independently screen titles, abstracts and full-text articles; and complete data extraction. For each study, we will extract data on the modality characteristics, primary outcomes (response rate and equivalence) and secondary outcomes (time, costs and acceptability to patients and providers). We will synthesise findings thematically without meta-analysis.

**Ethics and dissemination** Ethical approval is not required, as our review will include published and publicly accessible data. This review is part of a project to improve equitable access to eye care services in low-ioncome

## Strengths and limitations of this study

► As far as we are aware, this review will be the first to directly compare three commonly used data collection modalities for the collection of socioeconomic status data.
► The review will be comprehensive, covering published and grey literature in any language.
► This review will be robust, using independent dual review at every stage, and following best-practice guidelines.
► There may only be a small number of articles in the literature that compare the different modalities head to head and provide data on the outcomes of interest.

and middle-income countries. However, the findings will be useful to policy-makers and programme managers in a range of health settings and non-health settings. We will publish our findings in a peer-reviewed journal and develop an accessible summary of results for website posting and stakeholder meetings.

**PROSPERO registration number** CRD42021251959.

## INTRODUCTION
### Rationale

Inequalities in health are pervasive and stubbornly persistent. Individuals with lower levels of income, education and social status tend to experience the worst health outcomes irrespective of where they are in the world.[1] Tudor Hart observed that the availability of good medical care tends to vary inversely with the need for it in the population served.[2] This

inverse care law manifests in the majority of global health and development programmes where individuals with the lowest socioeconomic status (SES) tend to face the highest barriers in accessing care and are the least likely to attain good outcomes.

Recognising marked international and intranational disparities in health outcomes, the WHO was constituted in 1948 with the mandate of advancing 'health for all'.[3] The contemporary manifestation of this mission is encapsulated in the concept of Universal Health Coverage (and Sustainable Development Goal target 3.8,[4] which seeks to extend coverage to disenfranchised groups. Emerging emphases on attaining effective coverage,[5] and equitable coverage[6 7] seek to shift the success criteria from supply-side provision of services to demand-side receipt of effective services according to need. These trends are underpinned by the principle of 'proportionate universalism': seeking to improve the health of all, with the greatest gains experienced by those with the greatest needs.[8] There is also an increasing interest in understanding the distribution of programme benefits across sociodemographic groups—for instance women, those living in rural locations and those living in conditions of poverty.[9]

All attempts to boost equity in service provision are predicated on adequate collection and analysis of sociodemographic data. Previous work has demonstrated that sociodemographic data can be collected using a variety of modalities in the community setting including in-person, telephone voice calls and using automated telephone-based systems[10] (box 1). However, as far as we are aware, the relative costs and benefits of the different modalities have not been studied, including the skills, equipment, time and financial resources required and acceptability to data collectors and service beneficiaries.

This review aims to answer the research question 'how do three common SES data collection modalities compare in terms of performance characteristics, resource requirements and acceptability to participants and service providers?' Selecting an appropriate and cost-effective modality is an important first step towards advancing equitable effective service coverage.

The findings of this review will directly inform the development of school and community-based eye health screening programmes that operates in several low-income and middle-income countries (LMICs) including Botswana, Ethiopia, Kenya, Nepal, Pakistan, Tanzania, Uganda and Zimbabwe.[11] However, the collection of SES data is relevant for a much wider range of global health programmes, as well as non-health programmes aimed at improving educational, agricultural, gender equity and economic outcomes, among others.

## Descriptions of the interventions

Three different modalities for SES data collection constitute the interventions of interest for this review: in-person; voice call; and automated telephone data collection. Box 1 provides the definition for each.

---

| Box 1 | Definitions of the three data collection approaches used in this review |

In-person data collection includes any form of exchange between a programme implementer and a participant or their responsible guardian, whereby the programme implementer asks predefined questions to ascertain the participants' socioeconomic status (SES) and a synchronous response is received i.e, both parties occupy the same time and space, and the response is recorded by the implementer before the encounter is terminated. Any recording modality used by the programme implementer will be included, such as pen and paper or completion of an electronic form. For this review, we will also include self-administered questionnaires as a subtype of in-person data collection, provided that; the data collection instrument is provided when the participant presents to a programme implementer in-person; the participant is asked to complete the data entry form; and the participant submits their responses before departing. Any non-hospital location will be accepted.

Voice call data collection includes real-time, telephone-based verbal exchanges between programme implementers and participants whereby SES data are elicited and recorded by the programme implementer using predefined questions. This category includes computer-assisted telephone interviews (CATI)—where the interviewer follows prompts on a computer screen—as well as non-CATI. Videocalls will be included as a subtype of voice-calls.

Automated telephone-based data collection includes any mobile-telephone-based asynchronous exchange of information whereby participants are sent a standardised text message, multimedia message or automated phone call (sometimes called interactive voice response or 'IVR') and asked to provide SES data. Responses can be provided using the same modality or any other digital form for example, entering details on a webpage. Interventions that require participants to engage with human programme implementers will be excluded. All forms of phrasing of the requests and responses will be included. We will exclude data collection approaches that require the download of third-party software, including email. For this review we will include web-surveys that can be accessed by a hyperlink, reasoning that all smartphones come with a preloaded browser.

## Other terminology used in this review
### Community-based health programmes

For the purpose of this review, health programmes are defined as organised activities to improve one or more health outcome(s) in a defined population. Community-based care encompasses all settings except hospitals. Other definitions of community-based care exclude primary care facilities,[12] but these will be included in this review, along with outreach/mobile clinics, community centres, schools, workplaces and people's own homes.

### Programme implementers

Anyone with a formal responsibility to collect data on behalf of the health programme will be dubbed a 'programme implementer' for the purpose of this review. This term will cover voluntary and paid staff, and all cadre types.

### Participants

Any health programme beneficiary/recipient/client/patient that is asked to provide their SES data will be dubbed a 'participant' for the purpose of this review.

## Socioeconomic status

SES is a critically important but nebulous concept that pertains to social and economic standing within society.[13] It determines exposure to the social determinants of health; 'the conditions in which people are born, grow, live, work and age',[14] and relates to issues of privilege, power and control.[15] Almost all health outcomes are patterned according to SES, with the most disadvantaged populations experiencing the worst health outcomes.[13 16 17] SES is commonly measured using income, education, occupation and other metrics such as wealth, caste and place of residence. We will include all of these domains, as well as any other proxies that are identified by researchers as capturing SES.

## Low-income and middle-income countries

Just as health inequalities exist within countries—driven by differential access to resources, power, privilege and control—the same set of factors drive international health inequalities. Preston found that national life expectancy was tightly correlated with gross domestic product (GDP) by purchasing power parity, following a logarithmic path whereby small rises in GDP are initially associated with large gains in life expectancy, followed by increasingly diminishing returns.[18 19] In 1978, the World Bank first divided countries into 'low-income' and 'middle-income' groupings, based on gross national income (GNI) per capita. Whereas GDP captures the total value produced in a nation, GNI also includes net income received from overseas. Despite the fact that national finances are a fairly crude proxy,[20 21] many development agencies have come to use the World Bank categorisations to define eligibility for support. This review will use the World Bank analytic classifications for fiscal year 2021; defining LMICs as countries with GNI per capita ≤ US$12 535[22] using the Atlas method.[23]

## Objectives

We aim to systematically review the findings of empirical studies that have compared at least two different modalities for gathering SES data for community-based health programmes in terms of their resource requirements, performance characteristics, and acceptability to participants and service providers. Our findings should help programme managers make evidence-informed decisions when selecting the most appropriate modality for SES data collection.

## METHODS AND ANALYSES

This protocol is reported according to the relevant sections of the Preferred Reporting Items for Systematic Reviews and Meta-Analyses (PRISMA) Protocols guidelines.[24]

## Population

For this methodological paper, the 'population' is composed of studies rather than people, namely those that seek to compare two or more modalities for socioeconomic data collection from individuals enrolled in health programmes. Studies that only report on only one mode of data collection will be excluded. Studies conducted in hospital-based ambulatory care facilities will be excluded.

## Interventions

The interventions being studied are three different modalities for collecting socioeconomic data. The focus is on the modality of data collection (eg, in-person vs voice call vs automated) rather than the content of the wording that is used to elicit information.

Three different modalities for SES data collection constitute the interventions of interests for this review: in-person, voice-call and automated telephone systems, as defined in box 1. We will exclude approaches that use a blend of modes to elicit SES data. We will also exclude studies where the SES questions and wording are not kept constant across modes, for example, if a study asks about education via phone and face to face, the question must be worded in the same way for both approaches. This ensures that differences in response rates and other outcomes are only due to differences in mode of elicitation.

Studies that gather SES data at the household or community level will only be included if these data are used to make assumptions about the SES of identifiable individual participants enrolled (or due to be enrolled) in the service delivery programme of interest. Any two or more modalities can be studied. There is no index/ gold-standard data collection modality. Interventions that bundle requests for SES data with requests for other data (eg, broader demographic data) will be included, as long as separate results are reported for the SES data collection element. Interventions that use a blend of two or more modalities to request or receive data will be excluded. Studies that use email for data collection will be excluded.

## Comparator

In-person, voice call and automated telephone-based system attributes will be compared against each other. We will not include studies that only report outcomes for one modality i.e. where comparisons are not possible. For each mode, we will code the subtype of data collection, for example, distinguishing between computer-assisted telephone interviews (CATI) and non-CATI. There is a risk that response rates will be influenced by other items in the survey, setting and population. As such, our analysis will focus on outcome ratios between modes that pose the same questions in the same populations—rather than absolute levels as these may not be generalisable. We will report the wider context for each included study, and flag studies where SES questions are embedded within broader surveys that focus on taboo areas, for example, sexual behaviours or drug and alcohol use.

We will present outcomes for individual SES questions. We will only present data on identical questions

asked using different modes that is, if the wording is non-identical we will exclude the comparison from our analysis.

## Primary outcomes

There are two groups of primary outcomes; performance characteristics and resource requirements. We will report these at the level of individual SES items.

### Performance characteristics

► Response rate: number of completed SES items divided by the total number of elicitation attempts. This will be calculated at the level of each individual SES item.
► Equivalence: agreement between the responses obtained from two or more different modalities. Recognising that equivalence can vary by question, we will report equivalence for each individual SES item. We will report equivalence figures if they aggregate multiple SES questions in a secondary analysis, however, we will not report aggregate equivalence figures that mix SES items with non-SES items.
► Following Belisario and colleagues' Cochrane review,[25] we will use comparisons of mean scores between modalities and/or correlations and/or measures of agreement—which include intraclass correlation (ICC) coefficients, Pearson product–moment correlations, Spearman's r and weighted kappa coefficients.

### Resource requirements

► Time: the time taken to gather SES data using each approach (range and mean).
► Costs: any financial data on the costs of operating the data collection approach will be included. Fixed costs include the costs of equipment, software, insurance and personnel required to set up a given data elicitation modality. We will also include any ongoing support costs. We will aim to calculate the fixed and per-person costs to purchasers.

## Secondary outcome

### Acceptability to participants and service providers

Survey or interview results reporting on how programme implementers and participants feel about the data collection modality in terms of intrusiveness, ease of use, time requirement and general acceptability, as well as perceived advantages, barriers, disadvantages and additional costs presented by the beneficiaries, data collectors or study authors. This includes an assessment of socioeconomic barriers to accessing the modalities.

## Study types to be included

All empirical study designs that compare two or more data collection modalities will be included, for instance, in-person versus SMS approaches (SMS stands for 'short message service'). Studies must compare modalities that have been used to gather data from participants. Studies that use simulated data, or data obtained from populations other than the intended beneficiaries will be excluded. Both quantitative and qualitative study designs will be included as long as they report on one or more of the outcomes of interest. Review articles will not be included, but their primary studies will be screened for potential inclusion.

## Search methods for identification of studies

### Search strategy

The search strategy will be built around three blocks: the three data collection modalities, SES concepts and study design or study setting terms. The search will be limited to human studies published since 1999: the year that it first became possible to send cross-network SMS messages. We will search for full-text studies published in any language. We will not include reports of studies published as conference abstracts. The full search strategies used for each database are presented in the online supplemental appendix. The search will be performed on 29 June 2021. We plan to complete the review by October 2022.

### Electronic databases

We will search the following information resources: the Cochrane Library, MEDLINE, Embase and Global Health. We will search ClinicalTrials.gov and the WHO International Clinical Trials Registry Platform (ICTRP) for current and ongoing trials. OpenGrey will be searched for grey literature. The first 20 pages of Google Scholar will also be screened. We will check the reference lists of included studies and relevant systematic reviews to identify any additional potentially relevant reports of studies. Key authors will be contacted to uncover additional or upcoming studies.

### Measures of effect

We will calculate mean differences for methodological performance between the modalities, as well as for time and cost differences. For equivalence, we will follow Belisario et al[25] and Gwaltney et al,[26] using comparisons of mean scores between modalities and/or correlations and/or measures of agreement—which include ICC coefficients, Pearson product–moment correlations, Spearman's r and weighted kappa coefficients.

## Data collection and analysis

### Selection of studies

Initial screening of studies will be based on the information contained in their titles and abstracts, using online software (Covidence systematic review software, Veritas Health Innovation, Melbourne, Australia. Available at www.covidence.org). Studies that clearly do not meet the inclusion criteria will be excluded. The first 10% of papers will be screened by two reviewers collaboratively to align interpretation of the inclusion criteria and clarify the wording as appropriate. Any changes or amendments will be recorded. All remaining records will be screened independently by two reviewers. They will meet after every 10% batch of papers has been screened to discuss any issues. Any disagreements will be resolved through

consensus-based discussion, or if necessary, discussion with a third reviewer.

We will obtain full texts for the potentially relevant papers. Two review authors will independently assess the papers against the inclusion criteria to determine their eligibility for inclusion. Non-English language papers will be translated into English. The review authors will resolve disagreements through consensus-based discussion, or if necessary, discussion with a third reviewer. The reviewers will record reasons for exclusion at the full-text screening stage. A PRISMA flow diagram will be completed to summarise the study selection process.[27]

### Data extraction and management

Two review authors will independently extract study characteristics and data from the included studies using a custom Google Sheets data extraction form based on the Cochrane template for Randomised Controlled Trials (RCTs) and non RCTs.[28] The data extraction form will be piloted on 30 studies by two review authors and required amendments will be made by consensus. We anticipate a broad scope of included studies, so data charting will be an iterative process throughout the review, with agreement calculated and discussed at regular intervals (after each 10% batch of studies) and the data extraction form will be amended as required. Any discrepancies will be resolved by discussion, and a third reviewer will be consulted if necessary.

The following data will be extracted:
► Article title.
► Journal title.
► Authors.
► Country.
► Language.
► Publication year.
► Type of study.
► Focus of the service delivery programme.
► Sociodemographic characteristics for the population served: age, sex, urban/rural, ethnicity, marital status.
► Number of participants.
► Questions used to assess SES.
► Number of times SES data are collected from each participant.
► Types of intervention, including:
  – Modality.
  – Who gathers the SES data.
  – When in the patient journey/programme.
  – Equipment used.
  – Who provides the data.
  – Whether data collection is synchronous or asynchronous.
► Whether continuous improvement methods are used to refine the data collection approach, based on performance data.
► Types of comparison.
► Types of outcome measures.
► Outcomes: response rate, completeness, equivalence, time and costs—as described above.

► We will also extract all qualitative text provided on acceptability.

### Risk of bias assessment for included studies

We will use the Cochrane 'RoB2' tool for randomised studies[29 30] and 'RoB-I' for non-randomised studies.[31] Two reviewers will independently assess risk of bias. The review authors will resolve disagreements through a consensus-based decision, or if necessary, discussion with a third reviewer.

The risk of bias for each outcome across individual studies will be summarised as a narrative statement and supported by a risk of bias table. A review-level narrative summary of the risk of bias will also be provided.

### Contacting study authors

We will contact study authors to request additional information and primary data where any aspect precludes the assessment of eligibility or inclusion in the data synthesis.

### Strategy for data synthesis

If data are available, we will pool effect estimates using a random-effects model.[32] However, we anticipate heterogeneity in study design, interventions and outcomes and therefore plan to use a narrative 'synthesis without meta-analysis' approach, following the 'SWiM' reporting guidelines from Campbell *et al*.[33] We will stratify the synthesis by intervention type and outcome. Studies found to be at high risk of bias will be excluded from the synthesis.

### Assessment of heterogeneity

We will assess heterogeneity by considering study design, interventions and outcomes.

### Analysis of subgroups or subsets

We will assess whether response rates for each modality vary according to age, sex, urban/rural, ethnicity and marital status where baseline data on the distribution of these characteristics within the general population are available.

We will perform secondary analyses to examine whether findings differ between high-income and LMICs, and including all studies found to be at high risk of bias.

### Meta-biases

It is unlikely that we will be able to assess publication bias because it would require meta-analyses of 10 or more studies, but if we do have such an analysis we will create a funnel plot.[34] Selective outcome reporting will be assessed by comparing protocols (where available) with published reports.

### Assessment of certainty of evidence

Where possible, the GRADE criteria will be used to assess the certainty of the primary outcomes.[35 36] One review author will collate the evidence for each primary outcome and suggest initial ratings. These will be deliberated by a team of review authors who will reach a joint decision for each outcome. For RCTs, evidence will be assumed to be

high certainty and then will be downgraded due to risk of bias, inconsistency of results, indirectness of evidence, imprecision, publication bias. For observational studies, evidence starts at low-certainty but can be upgraded if there is a large effect, dose-response, gradient or plausible confounding that decreases the magnitude of effect.

## Patient and public involvement

No patient involved.

## Ethics and dissemination

Ethical approval is not required, as our review will only include published and publicly accessible data.

We will publish our findings in an open-access, peer-reviewed journal and develop an accessible summary of the results for website posting and stakeholder meetings. Data generated from this review will be made available on reasonable request.

**Author affiliations**
¹Department of Clinical Research, London School of Hygiene & Tropical Medicine, London, UK
²Institute of health and wellbeing, University of Glasgow, Glasgow, UK
³London School of Hygiene and Tropical Medicine International Centre for Eye Health, London, UK
⁴Department of Ophthalmology, University of Nairobi, Nairobi, Kenya
⁵Operation Eyesight, Nairobi, Kenya
⁶Peek Vision, London, UK
⁷London School of Hygiene & Tropical Medicine, London, UK
⁸London School of Hygiene and Tropical Medicine Faculty of Infectious and Tropical Diseases, London, UK

**Contributors** LNA conceptualised and planned the study with SM, IG, DB, APM, MJB, DM, MK, JR and AB. IG and LNA designed the search terms. IG conducted the search. LNA and SM conducted screening, extraction and quality scoring. DB, APM, SG, APM, MJB, NB, DM, MK, JR and AB helped to analyse and interpret the initial findings. LNA wrote the first draft with SM. IG, DB, APM, SG, AM, MJB, NB, DM, MK, JR and AB critically revised iterations of the manuscript. All authors read and approved the final protocol.

**Funding** This work was supported by the Wellcome Trust and the NIHR; grant number 215633/Z/19/Z. For the purposes of open access, the authors have applied a CC-BY public copyright license to any author accepted manuscript version arising from this submission. The study was sponsored by the London School of Hygiene & Tropical Medicine.

**Disclaimer** The funders had no role in the development of the protocol and will not play any role in the execution of the systematic review.

**Competing interests** None declared.

**Patient and public involvement** Patients and/or the public were not involved in the design, or conduct, or reporting, or dissemination plans of this research.

**Patient consent for publication** Not applicable.

**Provenance and peer review** Not commissioned; externally peer reviewed.

**ORCID iDs**
Luke Nelson Allen http://orcid.org/0000-0003-2750-3575
David Blane http://orcid.org/0000-0002-3872-3621
Ana Patricia Marques http://orcid.org/0000-0001-8242-7021
Jacqueline Ramke http://orcid.org/0000-0002-5764-1306

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
