## [Reviewer comments · BMJ Open]

ARTICLE DETAILS

TITLE (PROVISIONAL)	Performance and resource requirements of in-person vs voice call vs automated telephone-based socioeconomic data collection modalities for community-based health programmes: a systematic review protocol
AUTHORS	Allen, Luke; Mackinnon, Shona; Gordon, Iris; Blane, David; Marques, Ana Patricia; Gichuhi, Stephen; Mwangi, Alice; Burton, Matthew J; Bolster, Nigel; Macleod, David; Kim, Min; Ramke, Jacqueline; Bastawrous, Andrew

VERSION 1 – REVIEW

REVIEWER	Patrick Wightman University of Arizona Arizona Health Sciences Center, Center for Population Science and Discovery
REVIEW RETURNED	21-Oct-2021

GENERAL COMMENTS	The study design is very clearly laid out and easy to follow. My only suggestion is to provide 1) an explanation why outcomes for "single approaches" are to be included only for a secondary analysis, and 2) a description of what this secondary analysis will look like. Given the value of the overall objective of the study this approach seems unnecessarily limiting. (If you were to establish some "rigorous" criteria to ensure comparability, giving appropriate single-approach studies equal consideration would be even better.) Also, BMJ guidelines for study protocols indicate that the "the dates of the study should be included in the manuscript", but I did not see anything to this effect in the paper.
--

REVIEWER	Catherine Chittleborough University of Adelaide, School of Public Health
REVIEW RETURNED	18-Jan-2022

GENERAL COMMENTS	The rationale for this systematic review is that if we are serious about improving equity in service provision, then we need to have adequate measures of socioeconomic status (SES). In health service environments where we need to be efficient in our data collection processes, we need to know the best indicators of SES, and modality for collecting those indicators, in terms of response rate, acceptability, and costs. The paper provides a detailed description of the methods of the review, although there are a few points of clarification required. Self-administered questionnaires will be combined with in-person data collection. However, writing an answer on paper may elicit a different response, or response rate, than having to answer an interviewer, and these differences may be hidden if these modes are
--

	combined. Does voice call data collection include both computer-assisted telephone interviewing (CATI) and telephone interviews that are not computer-assisted? CATI technology incorporates skips and prompts based on participants' responses. Telephone interviews that are not computer-assisted may be more similar to interviews conducted face-to-face. The "Objectives and Population" section states that studies that only report one modality are excluded. This is inconsistent with the "Study types to be included" section that states articles that present the outcomes for single approaches will also be included for a secondary analysis. It is also unclear what analysis will be conducted in the "comparisons between data collection approaches using the subset of included studies that only examine one approach" (p11) – what could be compared if studies only use one modality? Focus is on the modality of data collection (e.g. in-person vs voice call vs automated) rather than the content of the wording that is used to elicit information. Does this mean any differences in response rate, completeness, costs, etc. across modes may not actually be due to mode, they may be due to differences in question wording? It is unclear why automated email-based SES elicitation has not been built into the search strategy if that mode is going to be included in the analyses. Response rate may be influenced by the focus of the data collection that is not related to SES questions. Two questionnaires may have identical SES items collected via the same modality but have very different response rates because the focus of the remaining questions is on very different topics, or asked in very different settings or populations, so any differences in response rates may not be related to mode. Equivalence will be calculated for studies that compare two or more approaches, but the Population section states that studies that only report on one mode of data collection will be excluded. Therefore equivalence will be calculated for all studies, because studies are only included if they include two or more modalities? SES items may not necessarily be asked as an instrument, with an overall mean score. So why and how will equivalence be calculated for the whole questionnaire? Why wouldn't equivalence be calculated for each SES item? There is potential for high equivalence between modes on one SES indicator, but low equivalence between modes on a different SES indicator. So the mode chosen to be used in practice may be influenced by the indicators of SES that are important for that context. On a similar note, if I am interpreting the analytic approach as intended, I think the authors will examine the primary and secondary outcomes for all SES items as a whole. What if the SES items are different across the two modes being compared? Or will effects only be extracted in cases where the same items are asked in each modality? Is completeness (proportion of missing items) calculated for the
--	--

	whole questionnaire, or only for the SES items? The first mention of Belisario and Gwaltney (p8) should include references. The data extraction section mentions that the point at which SES data are being collected (start, mid-point, or end of programme) will be extracted. What about the point at which SES data are collected within the questionnaire? If the SES items are included in a longer questionnaire about other issues, whether the SES items are asked at the start or the end of the questionnaire may make a difference to response rates or accuracy of responses. In the data synthesis, whether it is narrative or using random effects models, will all studies be included, or only those that are not rated as high risk of bias? It doesn't make sense to assess a study as being high risk of bias, meaning we would have little confidence in the results, but then to include it in the synthesis of effects.
--	--

VERSION 1 – AUTHOR RESPONSE

Reviewer: 1

Dr. Patrick Wightman, University of Arizona Arizona Health Sciences Center

Comments to the Author:

The study design is very clearly laid out and easy to follow. My only suggestion is to provide 1) an explanation why outcomes for "single approaches" are to be included only for a secondary analysis, and 2) a description of what this secondary analysis will look like. Given the value of the overall objective of the study this approach seems unnecessarily limiting. (If you were to establish some "rigorous" criteria to ensure comparability, giving appropriate single-approach studies equal consideration would be even better.)

Our team discussed this in depth and decided to exclude all studies that do not compare two or more modalities. We have removed mention of secondary analyses from the methods section.

Also, BMJ guidelines for study protocols indicate that the "the dates of the study should be included in the manuscript", but I did not see anything to this effect in the paper.

We have added the start and likely completion dates as requested.

Reviewer: 2

Dr. Catherine Chittleborough, University of Adelaide

Comments to the Author:

The rationale for this systematic review is that if we are serious about improving equity in service provision, then we need to have adequate measures of socioeconomic status (SES). In health service environments where we need to be efficient in our data collection processes, we need to know the best indicators of SES, and modality for collecting those indicators, in terms of response rate, acceptability, and costs. The paper provides a detailed description of the methods of the review, although there are a few points of clarification required.

Self-administered questionnaires will be combined with in-person data collection. However, writing an

answer on paper may elicit a different response, or response rate, than having to answer an interviewer, and these differences may be hidden if these modes are combined.

To clarify, we will not combine approaches, and we will report separate outcomes for each assess the relative merits of each. For the reasons you suggest we are excluding studies that use a mix of modes to elicit SES data. We have updated the text to make this clearer.

Does voice call data collection include both computer-assisted telephone interviewing (CATI) and telephone interviews that are not computer-assisted? CATI technology incorporates skips and prompts based on participants' responses. Telephone interviews that are not computer-assisted may be more similar to interviews conducted face-to-face.

We plan to include both types of approach and code them appropriately. We have updated the text to clarify this.

"For each mode we will code the sub-type of data collection e.g. distinguishing between computer-assisted and non-computer assisted telephone interviews."

The "Objectives and Population" section states that studies that only report one modality are excluded. This is inconsistent with the "Study types to be included" section that states articles that present the outcomes for single approaches will also be included for a secondary analysis. It is also unclear what analysis will be conducted in the "comparisons between data collection approaches using the subset of included studies that only examine one approach" (p11) – what could be compared if studies only use one modality?

We have decided not to include studies that only examine a single approach, and updated the text accordingly.

Focus is on the modality of data collection (e.g. in-person vs voice call vs automated) rather than the content of the wording that is used to elicit information. Does this mean any differences in response rate, completeness, costs, etc. across modes may not actually be due to mode, they may be due to differences in question wording?

This is a very important source of potential bias. We are only including studies that compare modes that use exactly the same questions and wording. We have added a note to make this clear and explain why:

"We will exclude studies where the SES questions and wording are not kept constant across modes. This ensures that differences in response rates and other outcomes are only due to differences in mode of elicitation."

It is unclear why automated email-based SES elicitation has not been built into the search strategy if that mode is going to be included in the analyses.

We are planning to exclude approaches that require the download of third-party software, including email. We will accept web-surveys that can be accessed with a simple hyperlink. We have updated the methods to clarify:

"We will exclude data collection approaches that require the download of third-party software, including email. For this review we will include web-surveys that can be accessed by a hyperlink, reasoning that all smartphones come with a pre-loaded browser."

Response rate may be influenced by the focus of the data collection that is not related to SES questions. Two questionnaires may have identical SES items collected via the same modality but have very different response rates because the focus of the remaining questions is on very different topics, or asked in very different settings or populations, so any differences in response rates may not be related to mode.

We are aware of this and will make sure we flag it in the analysis of the actual review. We have added this issue to the protocol:

“There is a risk that response rates will be influenced by other items in the survey, setting, and population. As such, our analysis will focus on outcome ratios between modes that pose the same questions in the same populations - rather than absolute levels as these may not be generalisable. We will report the wider context for each included study, and flag studies where SES questions are embedded within broader surveys that focus on taboo areas e.g. sexual behaviours or drug and alcohol use.”

Equivalence will be calculated for studies that compare two or more approaches, but the Population section states that studies that only report on one mode of data collection will be excluded. Therefore equivalence will be calculated for all studies, because studies are only included if they include two or more modalities?

We have cleared this up by deleting the offending text. You are correct that single-mode studies will be excluded. Equivalence will be calculated for all studies.

SES items may not necessarily be asked as an instrument, with an overall mean score. So why and how will equivalence be calculated for the whole questionnaire? Why wouldn't equivalence be calculated for each SES item? There is potential for high equivalence between modes on one SES indicator, but low equivalence between modes on a different SES indicator. So the mode chosen to be used in practice may be influenced by the indicators of SES that are important for that context.

This is a great point and we have changed our approach. We will only present equivalence when it is reported for individual SES questions. We have added this clarifying text:

“Recognising that equivalence can vary by question, we will report equivalence for each individual SES item. We will not report equivalence figures if they aggregate multiple questions.”

On a similar note, if I am interpreting the analytic approach as intended, I think the authors will examine the primary and secondary outcomes for all SES items as a whole. What if the SES items are different across the two modes being compared? Or will effects only be extracted in cases where the same items are asked in each modality?

We will present outcomes for individual SES questions. We will only present data on identical questions asked using different modes. We have clarified this in the text:

“We will present outcomes for individual SES questions. We will only present data on identical questions asked using different modes.”

Is completeness (proportion of missing items) calculated for the whole questionnaire, or only for the SES items?

We have added the following clarification:

“Completeness will be reported for individual SES questions, rather than for the whole questionnaire.”

The first mention of Belisario and Gwaltney (p8) should include references.

We have added this.

The data extraction section mentions that the point at which SES data are being collected (start, mid-point, or end of programme) will be extracted. What about the point at which SES data are collected within the questionnaire? If the SES items are included in a longer questionnaire about other issues, whether the SES items are asked at the start or the end of the questionnaire may make a difference to response rates or accuracy of responses.

We had not planned to collect this data on the basis that very few studies include the relevant details and it will affect both modes equally.

In the data synthesis, whether it is narrative or using random effects models, will all studies be included, or only those that are not rated as high risk of bias? It doesn't make sense to assess a study as being high risk of bias, meaning we would have little confidence in the results, but then to include it in the synthesis of effects.

We will exclude studies found to be at high risk of bias. We have added this information:

“Studies found to be at high risk of bias will be excluded from the synthesis.”

VERSION 2 – REVIEW

REVIEWER	Patrick Wightman University of Arizona Arizona Health Sciences Center, Center for Population Science and Discovery
REVIEW RETURNED	15-Mar-2022

GENERAL COMMENTS	All comments addressed
------------------------

REVIEWER	Catherine Chittleborough University of Adelaide, School of Public Health
REVIEW RETURNED	06-Mar-2022

GENERAL COMMENTS	The authors have addressed most of the queries and concerns that I raised previously. There are a couple of points that still remain to be clarified. Now that completeness and response rate will be calculated for individual SES items, it is not clear how completeness and response rate are different things. They both appear to be proportion of missing SES items. If they are different outcome measures, this needs to be clarified. The authors state in their response that they will exclude studies
---

	found to be at high risk of bias and stated that they added "Studies found to be at high risk of bias will be excluded from the synthesis". However the manuscript states that studies found to be at high risk of bias will be excluded from the data synthesis only in a secondary analysis. Why wouldn't the analysis that excludes the high risk of bias studies be the main analysis? How confident can we be in the findings if the main analysis is made up of a lot of studies at high risk of bias?
--	---

VERSION 2 – AUTHOR RESPONSE

Reviewer: 1

Dr. Patrick Wightman, University of Arizona Arizona Health Sciences Center

Comments to the Author:

All comments addressed

Reviewer: 2

Dr. Catherine Chittleborough, University of Adelaide

Comments to the Author:

The authors have addressed most of the queries and concerns that I raised previously. There are a couple of points that still remain to be clarified.

Now that completeness and response rate will be calculated for individual SES items, it is not clear how completeness and response rate are different things. They both appear to be proportion of missing SES items. If they are different outcome measures, this needs to be clarified.

The reviewer is correct in that both measure the same thing. We have removed all text pertaining to 'completeness' and added a short line of text to further clarify our approach in the methods:

- **Response rate:** number of completed SES items divided by the total number of elicitation attempts. This will be calculated at the level of each individual SES item.

The authors state in their response that they will exclude studies found to be at high risk of bias and stated that they added "Studies found to be at high risk of bias will be excluded from the synthesis". However the manuscript states that studies found to be at high risk of bias will be excluded from the data synthesis only in a secondary analysis. Why wouldn't the analysis that excludes the high risk of bias studies be the main analysis? How confident can we be in the findings if the main analysis is made up of a lot of studies at high risk of bias?

We have revised the protocol to exclude studies at high risk of bias from the synthesis. We will include all studies in a secondary analysis.

VERSION 3 – REVIEW

REVIEWER	Catherine Chittleborough University of Adelaide, School of Public Health
REVIEW RETURNED	22-Mar-2022
GENERAL COMMENTS	Thank you, the authors have addressed all of my queries and concerns.